# Is Dual-Task Training Clinically Beneficial to Improve Balance and Executive Function in Community-Dwelling Older Adults with a History of Falls?

**DOI:** 10.3390/ijerph191610198

**Published:** 2022-08-17

**Authors:** Jin-Hyuck Park

**Affiliations:** Department of Occupational Therapy, College of Medical Science, Soonchunhyang University, Asan 31538, Korea; roophy@naver.com; Tel.: +82-43-530-4773

**Keywords:** dual-task training, balance, executive function, cognitive training, falls

## Abstract

Purpose: To date, the effects of dual-task training on balance underlying cognitive function remain unclear. Therefore, this study was to verify the effects of cognitive–physical dual-task training on balance and executive function in community-dwelling older adults with a history of falls. Method: Fifty-eight participants were randomly allocated to the experimental group (EG) receiving cognitive–physical dual-task training (n = 29) or to the control group (CG) receiving functional balance training (n = 29). After 12 sessions for 6 weeks, the One Leg Standing Test (OLST), the Timed UP and Go (TUG), and part B of the Trail-Making Test (TMT-B) were implemented to examine static and dynamic balance and executive function. Results: After the 12 sessions, the EG showed a greater improvement in the OLST (*p* < 0.001; η^2^ = 0.332), the TUG (*p* < 0.001; η^2^ = 0.375), and the TMT-B (*p* < 0.001; η^2^ = 0.224) compared to the CG. Conclusion: These results indicate that dual-task training is clinically beneficial to improving static and dynamic balance as well as executive function in older adults with a history of falls. These findings shed new light on a clinical implication that executive function should be considered in balance training for older adults.

## 1. Introduction

Balance is an ability to properly control posture to adapt to various environments through the interaction of sensation, motor, and cognitive systems, which is significantly correlated with falls [1]. Specially, imbalance commonly occurs under dual-task conditions among older adults since cognitive function decreases with aging [2].

Dual-task refers to the ability to perform two or more cognitive and physical activities simultaneously [3]. Since both cognitive and physical functions decrease with aging, dual-task performance could deteriorate, resulting in falls in older adults while performing activities of daily living that require maintaining balance under dual-task conditions [4]. Falls result in fracture, concern about falls, and considerable morbidity, posing a major threat to the quality of life of older adults [5]. Therefore, treatments for reducing fall risks under dual-task conditions have gained a lot of attention [6,7].

A number of studies have identified the effects of dual-task training on improving balance and reducing fall risks in older adults, who are at a high risk of falls, such as people with stroke, Parkinson’s disease, or history of falls [2,8,9,10]. In most of the previous studies, dual-task training was conducted by performing cognitive tasks, such as word speaking or counting, while standing or walking on a treadmill or over the ground [3].

Although previous studies have reported positive effects of dual-task training, most of them explained its efficacy in balance in terms of improvements in physical components, such as velocity, step, and stride length, rather than cognitive components [6,11,12]. However, since cognitive factors significantly affect balance as well as physical components, the cognitive function underlying balance also needs to be investigated after dual-task training [3]. Indeed, recent studies have reported that high-level cognitive functions, such as attention shifting between movements and environments and inhibition of distracting stimuli, are necessary to maintain balance in complex environments while conducting different tasks simultaneously [4,8,13]. Specifically, given that a previous study showed that executive function deficits in older adults are closely associated with balance impairments, the effects of dual-task training on balance need to be established by executive function [14]. 

On the other hand, most of the previous studies indicated improvements in dynamic balance only [6,11,12]. Balance can be classified into static balance that enables to maintain posture without movement and dynamic balance required by adjusting a center for mass during movement [15]. Considering static balance also plays a crucial role for the prevention of falls in older adults [15], and it needs to be investigated after balance training.

Therefore, the aim of this study was to investigate the effects of dual-task training on static and dynamic balance and executive function in older adults with a history of falls. This study hypothesized that dual-task training would show greater improvements in balance and executive function compared to single-task training after 12 training sessions in older adults who experienced falls.

## 2. Materials and Methods

### 2.1. Study Design

This study was a single-blind study, and subjects were randomly assigned into the experimental group or the control group using random numbers generated by Python computer language. This assignment was conducted by an experimenter who was unaware of the purpose of the present study. All assessors, who are occupational therapists with abundant clinical experience and have familiarity with the outcome assessments, were blinded to the group assignment. This study was conducted for 6 weeks, and subjects were assessed pre- and post-intervention. All subjects provided written informed consent according to the Declaration of Helsinki (2004). The current study was approved by the local Institutional Review Board registered at the Thai Clinical Trials Registry (ID: TCTR20210720006). 

### 2.2. Subjects

Between May 2021 and July 2021, subjects were recruited from local senior centers through a recruitment notice in Seoul. A total of 72 older adults were screened, and then 58 were finally selected (Figure 1). The inclusion criteria were as follows: (1) over 65 years of age, (2) those who have experienced falls in the last six months, (3) those who ambulate independently without any assistance devices, and (4) those who understand a simple instruction as confirmed by the Korean version of Mini-Mental Status Examination (≥24). The exclusion criteria were as follows: (1) those who have any neurological, orthopedic, or psychological disorders, (2) those who have visual or auditory impairments, and (3) those who have not participated in any programs for improving balance in the last six months. Exclusion criteria were confirmed through self-report with a promise that subjects only report the truth. The criteria were in accordance with a previous study [16].

The number of subjects was calculated using G*Power (Informer Technologies, Dusseldorf, Germany) [17]. With reference to a previous study [16], the effect size was set at 1.23, the α error at a probability of 0.05, and the power at 0.95, resulting in a minimum of 19 subjects required for each group.

### 2.3. Intervention

All sessions were conducted by one occupational therapist with seven years of clinical experience in local senior centers. All subjects carried out a 45 min training session, twice a week for 6 weeks, and they only received the training program, which is assigned to each group. All sessions were one-on-one with the subject. The duration and intensity of the training sessions were derived from previous studies reporting positive effects of dual-task training [2] and functional balance training [18] on balance in older adults, respectively. A five-minute warm-up period was given to all subjects before each training program. 

In the EG, subjects receiving the dual-task training were instructed to practice balance tasks while simultaneously conducting cognitive tasks and were asked to maintain attention to both balance and cognitive tasks at all times. Detailed examples of the dual-task training program are described in Table 1. In the CG, subjects conducted the balance training consisting of subprograms focusing on body stability, body stability combined with hand manipulation, body transport, and body transport combined with hand manipulation. Subjects in both groups completed all 12 sessions without missing, which resulted in a total of 12 sessions. Table 2 indicates detailed examples of the balance training program.

### 2.4. Measurement

To assess static balance, the One Leg Standing Test (OLST) was used. In the OLST, subjects were asked to put their hands on their hips and raise one leg from the floor with their eyes closed. The amount of time was measured from when the leg was raised until the leg was set back down on the floor using a stopwatch. The measurement was repeated for the other leg, and the two times were averaged. It has high test–retest reliability (r = 0.96) [19]. 

The Timed Up and Go (TUG) test was conducted to evaluate dynamic balance. In the TUG, subjects sitting on a chair with armrests were asked to make a round trip of 3 m and then sit on the chair. The amount of time for the trip was measured using a stopwatch. Its test–retest reliability is 0.96 [20].

To examine executive function, part B of the Trail-Making Test (TMT-B) was conducted. In the TMT-B, a sheet with 25 circle-shaped letters and numbers was presented, and subjects were instructed to connect the circles in alternating number–letter order. A maximum time of 300 s was given to complete it. The time to complete the task was recorded using a stopwatch [21].

### 2.5. Statistical Analysis

All data were analyzed using SPSS for Windows (version 22.0) (IBM, Armonk, NY, USA). The normal distribution of outcome variables was confirmed using the Shapiro–Wilk test. To compare subjects’ general characteristics between both groups, independent *t*-test and Chi-square analysis were used. 

After 12 training sessions, a repeated two-way analysis of variance (ANOVA) was implemented to compare differences in outcome measurements between both groups. The effect size (ES) of each training was calculated using partial η^2^ value. Partial η^2^ ≥ 0.14 was considered a large effect; between ≥0.06 and <0.14 a moderate effect; and between ≥0.01 and <0.06 a small effect [22]. The level of statistical significance was set at *p* < 0.05.

## 3. Results

### 3.1. Subject’s Characteristics

There were no significant differences in general characteristics between both groups (Table 3).

### 3.2. Balance

Repeated ANOVA showed that group × time interaction was significant for the OLST (*p* < 0.001; η^2^ = 0.332) and the TUG (*p* < 0.001; η^2^ = 0.375), indicating that subjects in the EG showed greater improvements in both static and dynamic balance compared with those in the CG (Table 4).

### 3.3. Executive Function

There was a significant group × time interaction for the TMT-B (*p* < 0.001; η^2^ = 0.224). This finding revealed that subjects in the EG achieved a greater improvement in executive function compared with those in the CG (Table 5).

## 4. Discussion

This study examined whether cognitive–physical dual-task training could be beneficial to improve balance and executive function in community-dwelling older adults with a history of falls. The findings of this study show that dual-task training is clinically beneficial to improving balance, which supports the hypothesis that balance could be enhanced by dual-task training.

In this study, all subjects in both the EG and the CG were asked to achieve postural control while conducting physical activities inducing postural sway, such as throwing a ball and carrying a bag. These activities involve complex interactions among somatosensory, visual, and vestibular systems controlling the relationships between body segments and external environment [1], which has a positive effect on balance supported by the improvements in static and dynamic balance in both groups. Indeed, a previous study indicated that plantar perception training as a somatosensory system exercise resulted in clinical improvements in static and dynamic balance [23]. Similarly, a previous study demonstrated that eyeball training as a vestibular system exercise enhanced balance [24]. Taken together, it was demonstrated that training using interactions among these systems could be a way to improve balance.

On the other hand, the findings of this study show that subjects in the EG achieved greater improvements in both static and dynamic balance compared to subjects in the CG. This result indicates that dual-task training might be more effective in improving balance than functional balance training, which is consistent with previous studies including the study referred to for calculating the number of subjects [16,25,26,27]. Previous studies reported that postural instability could be caused by attention deficits [28,29]. A previous study showed that attentional demands are closely associated with postural control in older adults [8]. With aging, inputs for vision and somatosensory were reduced due to high thresholds for the sensations [30]. Therefore, older adults require more attentional resources to maintain balance to compensate for decreases in sensory inputs [8]. Indeed, although sensory inputs were reduced, older adults with high attention capacity showed less postural sway than those with low attention capacity, suggesting cognitive components could play a crucial role in balance [8]. In this study, subjects in the EG were asked to control balance while simultaneously conducting cognitive tasks, which requires paying continuous attention to both tasks and inhibit interferences between cognitive and physical tasks, which is the core of executive function [31]. In other words, executive function is essential for successful dual-task training, supported by the results of this study that subjects in the EG showed improvement in the TMT-B, which adds a dimension of attention shifting, inhibition, and cognitive flexibility consisting of executive function compared to the TMT-A. Consequently, dual-task training induced positive effects on balance by enhancing executive function, which is in line with a previous study [14]. 

A balance impairment is one of the hallmarks of falls in older adults [32], and a growing body of evidence has reported the importance of cognitive factors to a balance impairment in older adults [33]. Thus, it is important to establish evidence on effective interventions to improve balance in older adults by enhancing cognitive function. This study implies that an enhancement in executive function could be a useful way to improve balance in older adults. The findings of this study have a clinical significance in that dual-task training could be performed to improve balance in the elderly who are not suitable for balance training focusing on physical components due to decreased flexibility and muscle strength. In addition, maintaining balance under dual-task conditions in older adults has more ecological validity compared to conventional balance training because a variety of activities of daily living involve simultaneous performance of multiple tasks challenging cognitive and motor capacities [6]. Therefore, dual-task training could be considered to be more effective than conventional balance training considering its ecological validity.

The present study suggests that dual-task training could be an alternative option instead of conventional balance training given that a cognitive factor might be important to reduce fall risk. Specifically, considering that community-dwelling older adults with a history of falls are at high risk for falls, dual-task training would be clinically beneficial to improve balance by improving executive function with its ecological validity in community settings. Nevertheless, this study has some limitations. First, no follow-up assessments were implemented, which limits knowledge of how long subjects in the EG were able to maintain improved balance and executive function. Second, this study did not investigate balance improvements in real environments. Since balance was assessed in the environment in which the training was carried out, the similarity of the environment might influence the present results. Third, although most of the previous studies have consistently reported that learning effects could not appear clearly with only two trials of outcome measurements [11,12,14], it is impossible to exclude them as this study did not involve inactive controls. Finally, neuroimaging devices were not used to observe the changes in the prefrontal cortex, which is mainly in charge of executive function [34], so this study could not confirm the possibility that dual-task training could be used for facilitating neuroplasticity of the prefrontal cortex. Therefore, future studies need to investigate the long-term effects of dual-task training on balance in a variety of real environments and executive function by using neuroimaging devices.

## 5. Conclusions

This study demonstrated that 12 training sessions of dual-task training are more helpful in improving balance in older adults with a history of falls compared with conventional balance training, suggesting the feasibility that dual-task training could be used as an alternative method to improve balance in older adults with its ecological validity in community settings. Moreover, this study implies that executive function could be considered as one of the factors for improving balance in older adults in everyday life and especially suggests that improving executive function is useful as balance training for physically frail older adults.

## Figures and Tables

**Figure 1 ijerph-19-10198-f001:**
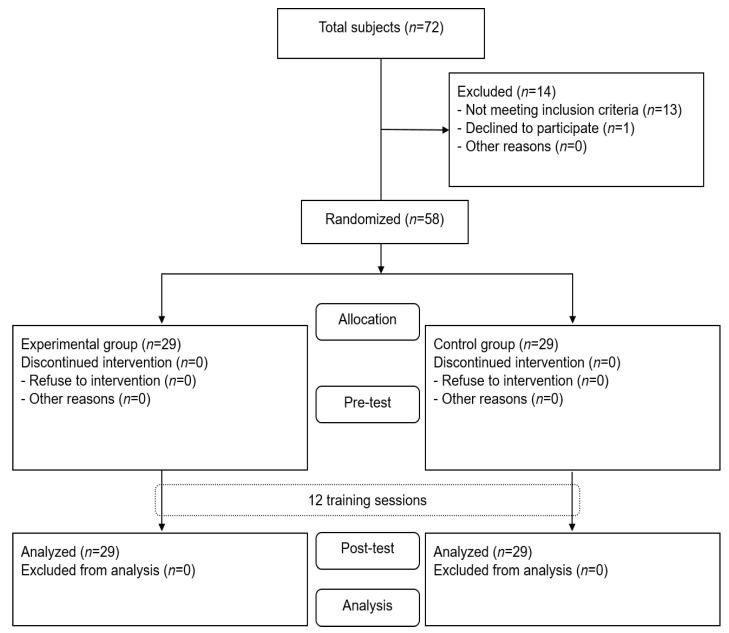
Flow diagram of subjects in the study.

**Table 1 ijerph-19-10198-t001:** Dual-task training program.

Programs	Contents	Periods (Minute)
Warm-up	- Stretching exercise	5
Dual-task training contents	- Placing each foot alternately on a step and decreasing UE support while spelling words backward	10
	- Continuously walking carrying a grocery bag while randomly naming numbers backward (double-digit)	10
	- Stepping over obstacle (height: 15 cm, distance: 30 cm) while reciting number, days, or months backward	10
	- With four standard armchairs placed at four corners of a square, repeatedly walking to the chair directly in front, sitting, then standing up, and walking to the chair on the left while calculating simple addition or subtraction (double-digit)	10

**Table 2 ijerph-19-10198-t002:** Balance training program.

Programs	Contents	Periods (Minute)
Warm-up	- Stretching exercise	5
Functional balance training contents	- Body stability (standing with eyes closed, tandem standing, and standing on various surfaces)	10
	- Body stability combined with hand manipulation (standing on a foam while throwing and catching a ball, tandem standing while holding a grocery bag)	10
	- Body transport (narrow walking, walking backward, transferring from one chair to another)	10
	- Body transport combined with hand manipulation (narrow walking while throwing and catching a ball, walking backward while holding a grocery bag)	10

**Table 3 ijerph-19-10198-t003:** General characteristics of participants (N = 58).

Characteristics	Dual-Task Group (n = 29)	Balance Training Group (n = 29)	χ^2^/t
Age, years (SD)	71.76 ± 3.14	70.97 ± 2.78	1.022
Height (cm)	158.61 ± 3.83	157.86 ± 4.07	0.670
Weight (kg)	61.70 ± 3.82	61.62 ± 3.47	0.602
K-MMSE	27.86 ± 1.24	28.03 ± 1.18	−0.540

MMSE-K: Mini-Mental Status Examination; SD: Standard deviation.

**Table 4 ijerph-19-10198-t004:** Changes in static and dynamic balance (N = 58).

Variables	Dual-Task Group (n = 29)	Balance Training Group (n = 29)	Between-Group Differences	*p*	η^2^
OLST (seconds)					
Pre-intervention	3.50 ± 0.48	3.47 ± 0.45	1.97 (1.22 to 2.71)	<0.001	0.332 ***
Post-intervention	7.52 ± 1.57	5.25 ± 1.57			
Within-group changes	3.74 (3.19 to 4.29)	1.77 (1.25 to 2.30)			
TUG test (seconds)					
Pre-intervention	13.81 ± 1.41	14.11 ± 1.48	1.64 (1.08 to 2.20)	<0.001	0.375 ***
Post-intervention	10.63 ± 1.11	12.57 ± 1.54			
Within-group changes	3.74 (3.19 to 4.29)	1.54 (1.25 to 1.82)			

(A 95% confidence interval) for within and between-group changes. OLST, One Leg Standing Test; TUG, Timed Up and Go test. *** Significant group × time interaction (*p* < 0.001).

**Table 5 ijerph-19-10198-t005:** Changes in static and dynamic balance (N = 58).

Variables	Dual-Task Group (n = 29)	Balance Training Group (n = 29)	Between-Group Differences	*p*	η^2^
TMT-B (seconds)					
Pre-intervention	89.98 ± 9.78	89.07 ± 10.82	1.55 (0.78 to 2.31)	<0.001	0.224 ***
Post-intervention	87.65 ± 10.00	88.29 ± 10.83			
Within-group changes	2.33 (1.64 to 3.02)	0.78 (0.39 to 1.17)			

(A 95% confidence interval) for within and between-group changes. TMT-B, Trail Making Test-B. *** Significant group × time interaction (*p* < 0.001).

## Data Availability

The data presented in this study are available on request from the corresponding author. The data are not publicly available because they are part of an ongoing project.

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
