# Peer review of "Is Dual-Task Training Clinically Beneficial to Improve Balance and Executive Function in Community-Dwelling Older Adults with a History of Falls?"

_ijerph, 2022, doi:10.3390/ijerph191610198_

Round 1
Reviewer 1 Report
Review report – 1851254 – IJERPH
A brief summary
Goal of the research was to verify the effects of cognitive-physical dual-task training on balance and executive function in community-dwelling older adults with a history of falls. Findings, according to authors, shed new light on a clinical implication that executive function should be considered in balance training for older adults.
Broad comments
Introducing overview was supported by methodology used to fulfill aim of the experiment. Experimental design was genneraly appropriate, followed and supported by relevant and precise conclusions. Limitations of the study were mostly correctly explained. However, one limitation of the experimental design and conclusion may present a flaw (in specific comments). If correctly disclosed, this paper presents a very nice contribution to interdisciplinary racourse of the subject.
Specific comments.
· Minor spell check, and style corrections.
· Ln 137-140: “Repeated ANOVA showed that group × time interaction was significant for the OLST (p < .001; η2 = .332) and the TUG (p < .001; η2 = .375), and TUG (p < .001; η2 = .174), indicating that subjects in the EG showed greater improvements in both static and dynamic balance compared with those in the CG (Table 4).” – TUG related results presented in non-coherent manner
· Ln 206-216: limitations which stem out of ‘learning effect’ on/of tests TUG, OLST, between ‘pre-‘ and ‘post-‘ measurements were not presented. It would be covered by control group without any intervention (and then with partialization of results in EG and CG). Environmental circumstances (with relation to exercise surroundings) is not sufficiently supporting argument. If not presented and supported (in literature if necessary), it may be a ‘major flaw’
· Sample size calculations within effect size from reference 16 was in line with findings? Please comment.
· Figure 1. Flow diagram is not intuitive (especially with relation to pre-test, post-test… it is not clear if it is for EG solely or…)
Author Response
1. Minor spell check, and style corrections.
: Thank you for your comment. I have reviewed to correct things.
2. Ln 137-140: “Repeated ANOVA showed that group × time interaction was significant for the OLST (p < .001; η2 = .332) and the TUG (p < .001; η2 = .375), and TUG (p < .001; η2 = .174), indicating that subjects in the EG showed greater improvements in both static and dynamic balance compared with those in the CG (Table 4).” – TUG related results presented in non-coherent manner
: Thank you for pointing this out. There was a mistake. I have corrected it according to the statistical analysis.
3. Ln 206-216: limitations which stem out of ‘learning effect’ on/of tests TUG, OLST, between ‘pre-‘ and ‘post-‘ measurements were not presented. It would be covered by control group without any intervention (and then with partialization of results in EG and CG). Environmental circumstances (with relation to exercise surroundings) is not sufficiently supporting argument. If not presented and supported (in literature if necessary), it may be a ‘major flaw’
: I totally agree with your comment. However, in most of previous clinical trials including relevant studies, it has been consistently suggested that learning effects could not appear clearly with just two measurements (‘pre- and post-measurements). Nevertheless, I have added this limitation as this study did not involve inactive controls. Thank you for your comments.
4. Sample size calculations within effect size from reference 16 was in line with findings? Please comment.
: The findings of this study were consistent with the #16 study. I have added it in the Discussion section. Thank you for pointing this.
5. Figure 1. Flow diagram is not intuitive (especially with relation to pre-test, post-test… it is not clear if it is for EG solely or…)
: I have revised figure 1 in accordance with the consort flow chart. Thank you for your comment.
Reviewer 2 Report
Thank you for the opportunity to review this well-written manuscript. I have specific recommendations regarding some outdated reference sources.
References:
Please make sure that all references are in the required format for Journal Articles. Most of your Journal Names are not italicized and the years are not bolded.
Author 1, A.B.; Author 2, C.D. Title of the article. Abbreviated Journal Name Year, Volume, page range.
I also have specific recommendations about the references.
1. A textbook is not an appropriate reference for a formal research paper. Textbooks are a secondary source. Please find the primary peer-reviewed source of this information.
The following references are outdated. Please work to find more recent information to support your points: 2. 4. 5 (MUST update), 8 (MUST update, this is an article on "emerging" area of research from 20 years ago), 10 (update as this was a "pilot" from 14 years ago, 18. 28. 29. 30. 31. 32.
Methods, Line 86: "All assessors were blinded to the group assignment." What was the clinical background of these assessors and how did you ensure they were trained in the procedures of the standardized assessments?
Line 72: Please provide more transparent detail about the recruitment methods. How was this done and over what time period?
Line 77: Exclusion criteria, how did you know this, was it self-reported by the participant?
Line 89: Were these sessions one-on-one with the participant, or were they provided in a group format? How many sessions could they miss and still be included in the final analysis? In Line 218 it states: "12 training sessions..." This seems to imply that there were no training sessions missed. Please include with the duration/dosing in this section, that this included 12 training sessions over 6 weeks, in alignment with the language used in Line 218.
Line 106: Would you be able to provide examples of all measures and their use with the older adult population?
Author Response
1. Please make sure that all references are in the required format for Journal Articles. Most of your Journal Names are not italicized and the years are not bolded.
Author 1, A.B.; Author 2, C.D. Title of the article. Abbreviated Journal Name Year, Volume, page range.
: I have reviewed the references to ensure that they are consistent with the journal format. Thank you for pointing it.
2. A textbook is not an appropriate reference for a formal research paper. Textbooks are a secondary source. Please find the primary peer-reviewed source of this information.
: I have replaced the textbook references into research papers.
3. The following references are outdated. Please work to find more recent information to support your points: 2. 4. 5 (MUST update), 8 (MUST update, this is an article on "emerging" area of research from 20 years ago), 10 (update as this was a "pilot" from 14 years ago, 18. 28. 29. 30. 31. 32.
: I have updated the references as you commented.
4. Methods, Line 86: "All assessors were blinded to the group assignment." What was the clinical background of these assessors and how did you ensure they were trained in the procedures of the standardized assessments?
: I have added the assessor’s information. They are all occupational therapists with abundant clinical experience and have familiarity with the assessments.
5. Line 72: Please provide more transparent detail about the recruitment methods. How was this done and over what time period?
: I have added the recruitment information in the Methods section.
6. Line 77: Exclusion criteria, how did you know this, was it self-reported by the participant?
: Thank you for your comment. It was confirmed through self-report with a promise that the subjects only report the truth. I have added this information.
7. Line 89: Were these sessions one-on-one with the participant, or were they provided in a group format? How many sessions could they miss and still be included in the final analysis? In Line 218 it states: "12 training sessions..." This seems to imply that there were no training sessions missed. Please include with the duration/dosing in this section, that this included 12 training sessions over 6 weeks, in alignment with the language used in Line 218.
: I have added the information about the intervention dose in detail. All subjects completed all 12 training sessions without missing.
8. Line 106: Would you be able to provide examples of all measures and their use with the older adult population?
: Intervention’ contents and outcome measures used in this study came from prior studies involving older adults.
Reviewer 3 Report
Researchers conducted an interesting study examining the effect of dual-task training on executive function. They found that dual-task training was effective in executive function. The study can be considered a pilot study. This is a valuable study into the importance of dual-task training.
On line 78 in the method section, the researchers wrote that individuals with hearing and vision problems were excluded. What method was used to determine whether the participants had the hearing and vision problems? Necessary explanations should be added in this regard.
Author Response
1. On line 78 in the method section, the researchers wrote that individuals with hearing and vision problems were excluded. What method was used to determine whether the participants had the hearing and vision problems? Necessary explanations should be added in this regard.
Thank you for your point. This point is exactly the same as what another reviewer indicated. The exclusion criteria were confirmed through self-report with a promise that the subjects only report the truth. I have added this information
Round 2
Reviewer 1 Report
All corrections are now included.
Recommendation proceeded to Editors.